# Characterization of a Novel Hyperthermophilic GH1 β-Glucosidase from *Acidilobus* sp. and Its Application in the Hydrolysis of Soybean Isoflavone Glycosides

**DOI:** 10.3390/microorganisms12030533

**Published:** 2024-03-07

**Authors:** Jinjian He, Yuying Li, Xihang Sun, Dinghui Zuo, Mansheng Wang, Xia Zheng, Pinglian Yu, Pengjun Shi

**Affiliations:** 1Institute of Bast Fiber Crops, Chinese Academy of Agricultural Sciences, Changsha 410205, China; hejinjian_@outlook.com (J.H.); liyuying01@caas.cn (Y.L.); sunxihangcn@163.com (X.S.); zbzdh42@163.com (D.Z.); wangmansheng@caas.cn (M.W.); zhengxia@caas.cn (X.Z.); 2College of Horticulture and Landscape, Tianjin Agricultural University, Tianjin 300392, China; 3Key Laboratory of Yunnan University for Plateau Characteristic Functional Food, School of Chemistry and Chemical Engineering, Zhaotong University, Zhaotong 657000, China

**Keywords:** β-glucosidase, *Acidilobus* sp., thermophilic, thermostability, ethanol tolerance, isoflavone glycosides

## Abstract

A putative β-glucosidase gene, *BglAc*, was amplified from *Acidilobus* sp. through metagenome database sampling from a hot spring in Yellowstone National Park. BglAc is composed of 485 amino acid residues and bioinformatics analysis showed that it belongs to the GH1 family of β-glucosidases. The gene was successfully expressed in *Escherichia coli* with a molecular weight of approximately 55.3 kDa. The purified recombinant enzyme showed the maximum activity using *p*-nitrophenyl-β-D-glucopyranoside (*p*NPG) as the substrate at optimal pH 5.0 and 100 °C. BglAc exhibited extraordinary thermostability, and its half-life at 90 °C was 6 h. The specific activity, *K*_m_, *V*_max_, and *K*_cat_/*K*_m_ of BglAc toward *p*NPG were 357.62 U mg^−1^, 3.41 mM, 474.0 μmol min^−1^·mg^−1^, and 122.7 s^−1^mM^−1^. BglAc exhibited the characteristic of glucose tolerance, and the inhibition constant *K*i was 180.0 mM. Furthermore, a significant ethanol tolerance was observed, retaining 96% relative activity at 10% ethanol, and even 78% at 20% ethanol, suggesting BglAc as a promising enzyme for cellulose saccharification. BglAc also had a strong ability to convert the major soybean isoflavone glycosides (daidzin, genistin, and glycitin) into their corresponding aglycones. Overall, BglAc was actually a new β-glucosidase with excellent thermostability, ethanol tolerance, and glycoside hydrolysis ability, indicating its wide prospects for applications in the food industry, animal feed, and lignocellulosic biomass degradation.

## 1. Introduction

β-glucosidase (EC 3.2.1.21) is a widespread group of glucoside hydrolase that catalyzes the hydrolysis of glycosidic linkages between two or more carbohydrates, or between a carbohydrate and a non-carbohydrate moiety, to release glucose or shorter oligosaccharides [1]. Glycoside hydrolases have been investigated in the past years due to their significant roles in industrial applications, such as the hydrolysis of soybean isoflavone glucosides [1,2,3] and rare ginsenosides [4,5,6] and the conversion of biomass to fuel ethanol [7,8,9,10]. Particularly, the hydrolysis of soybean isoflavone glycosides could bring significant economic value. So far, β-glucosidases used on the hydrolysis of soybean isoflavone glycosides have been studied in depth using *Thermotoga maritime* [11], *Thermoanaerobacter ethanolicus* [12], *Pyrococcus furiosus* [13], *Neosartorya fischeri* [14], *Talaromyce leycettanus* [15], *Alicyclobacillus* sp. [16], and *Aspergillus niger* [3].

β-glucosidases widely exist in all domains of living organisms, like in archaea, bacteria, fungi, insects, plants, and animals [2]. β-glucosidases are classified into 173 families according to the homology of amino acid sequences [17] (http://www.cazy.org/ 6 January 2024). Because the fold of proteins is better conserved than their sequences, some families can be grouped into ‘clans’ based on their proteins. Of these, clan GH-A has the largest number of families. The GH1 family is also grouped within GH-A, which contains 26 families and more than 120 family members (http://www.cazy.org/ 6 January 2024). Prior to this study, the crystal structures of more than four glucosidases were resolved, and it was demonstrated that the 3D structure of glycosidase GH1 is composed of a (β/α)_8_ barrel [1,18]. The β-glucosidase investigated in this study is derived from the archaea *Acidilobus* sp., which belongs to the first glycosidase family (GH1). Archaea are the oldest microorganisms on the earth, showing adaptability to extreme environments, such as extreme acidophilic, alkaliphilic, psychrophilic, and thermophilic environments. In particular, thermophilic and hyperthermophilic enzymes have broad application potential in industrial production. The hyperthermophilic glycoside hydrolases amplified from prokaryote archaebacteria such as *Pyrococcus furiosus* [19], *Sulfolobus solfataricus* [20], *Pyrococcus kodakaraensis* [21], *Thermoplasma acidophilum* [22], *Sulfolobus acidocaldarius* [23], *Acidilobus saccharovorans* [24], *Thermotoga naphthophila* [25], and *Thermococcus kodakarensis* [26] have been investigated, and their enzyme activities have been determined. The optimum temperature and pH range of the hyperthermophilic glucosidases characterized in this species were 78–105 °C and 5.0–7.0, respectively.

Enzymes with thermostable characteristics could tolerate harsh, high-temperature environments [24], as well as promote conversion efficiency and reduce costs in large-scale applications. However, the catalytic efficiency of β-glucosidases is affected by many factors such as thermostability, glucose, and pH. The way to obtain β-glucosidases which performed well for thermostability is to make mutants except to screen organisms living in extreme environments. The sequence and model protein structure of the β-glucosidases gene from *Acidilobus* sp. can be acquired from NCBI and PDB databases, facilitating genetic cloning to create mutants by rational design. This paper characterizes a novel β-glucosidases gene discovered from *Acidilobus* sp. and studies the protein expression, enzymatic characteristics, and amino acid sequence sites which affect the thermostability.

## 2. Materials and Methods

### 2.1. Materials and Chemicals

Unless specified otherwise, all chemicals were of analytical grade. Gentiobiose, Sophorose, 4-Nitrophenyl-β-D-xylopyranoside (*p*NPX), 4-Nitrophenyl-α-L-arabinofuranoside (*p*NPAf), and 4-Nitrophenyl-β-D-mannopyranoside (*p*NPman) were purchased from Megazyme (Bray, Ireland). Cellobioside, *p*-nitrophenyl-β-D-galactopyranoside (*p*NPGal), *p*-nitrophenyl-β-D-glucopyranoside (*p*NPG), o-nitrophenyl-β-D-galactopyranoside (*o*NPGal), daidzin, genistin, glycitin, daidzein, genistein, and glycitein were purchased from Sigma (St. Lois, MO, USA). The expression vector pET-30a (+), *E. coli* DH5α strain, and expression host strain *E. coli* BL21 (DE3) were purchased from Tiangen (Beijing, China). A 12% TGX Strain-Free FastCast Acrylamide Kit (Cat. # 1610185) was purchased from Bio-Rad (Hercules, CA, USA).

### 2.2. Gene Sequence Analysis and Cloning of BglAc from Acidilobus sp.

Multiple sequence alignment of BglAc and other similar protein were performed with the Clustal W software and depicted by GeneDoc. The 3D structure of BglAc was predicted using the Swiss-Model server (https://swissmodel.expasy.org/ 6 January 2024) with 4HA3 as the model, and the pictures were drawn using Pymol (https://pymol.org/2/ 6 January 2024).

The full-length sequence of BglAc (GenBank accession No. WYEC01000097) obtained from the genome of *Acidilobus* sp. [27] was synthesized by the Tsingke Biotechnology Company, Limited (Beijing, China). Then, the synthesized gene product was inserted into the pET-30a (+) vector using *Nde*I and *Eco*RI restriction sites to construct recombinant plasmid pET-30a-*BglAc* to transform into *E. coli* DH5α competent cells. Recombinant plasmid DNA was extracted from *E. coli* DH5α and verified by double digestion on 1.0% agarose gel.

### 2.3. Expression and Purification of Recombinant Protein BglAc

The recombinant plasmid pET-30a (+)-*BglAc* was transformed into the component cells of *E. coli* BL21 (DE3). The selected positive transformants were cultured in Luria–Bertani (LB) broth with kanamycin (50 mg/mL) at 37 °C overnight at 220 rpm in an oscillating incubator, then the seed culture was placed into new fresh LB broth in a 1000 mL flask with 50 mg/mL kanamycin. The ratio of bacterial liquid to the medium was 1:100. After incubating for almost 3 h, the absorbance of the culture at 600 nm wavelength reached 0.6 to 0.8, then isopropyl-β-D-thiogalactopyranoside (IPTG) was added to induce the protein expression of BglAc at a final concentration of 0.6 mM. The culture cells were continuously incubated at 30 °C for 4 h in an oscillating incubator at a speed of 220 rpm. After expression was induced, the bacteria culture was collected by centrifuging for 5 min at 10,000 rpm at 4 °C and resuspended in the sodium citrate buffer. The resuspended solution was ultrasonically crushed for 30 min at 4 °C and centrifugated at 10,000 rpm at 4 °C for 5 min to obtain the supernatant. The recombinant protein in this supernatant was purified using the Ni-NTA column according to the manufacturer’s handbook (Hilden, Germany). The purity of the recombinant protein was determined by 12% SDS-PAGE and the protein was quantified by the BCA Protein Assay Kit and using bovine serum albumin (BSA) as the standard.

### 2.4. Enzyme Activity Assay

*p*NPG was used as the substrate following the modified Craven method [28] to analyze the β-glucosidase activity. A quantity of 250 μL of the substrate solution (4 mM) was preheated at an appropriate temperature for 5 min, then 250 μL of the enzyme was added, and the reaction was initiated. Ten minutes later, the reaction was terminated with 1.5 mL of Na_2_CO_3_ solution (1 M). As a blank, Na_2_CO_3_ was added before the addition of the enzyme, and the enzyme was substituted with an equal amount of buffer solution. The enzyme activity was measured at 405 nm using a UV spectrophotometer. One unit (U) of enzyme activity was defined as the amount of enzyme required to release 1 μmol of *p*-nitrophenyl per min under the described conditions.

### 2.5. Optimum pH and Temperature of BglAc

To determine the effect of pH and temperature on β-glucosidase activity, pH values were adjusted to range from 3.0 to 10.0 with sodium citrate buffer (pH 3.0–8.0) and tris–HCl buffer (pH 8.0–10.0). Likewise, the effect of temperature on enzyme activity was determined at a pH of 5.0 and temperatures ranging from 50 °C to 100 °C.

### 2.6. Thermostability Determination

The thermostability of BglAc was assayed by analyzing the relative activity after preincubating at optimal temperatures (100 °C and 90 °C) and sampling every hour or every ten minutes. The enzyme activity was measured by the same method mentioned above.

### 2.7. Effects of Metal Ions and Chemical Reagents on Enzyme Activity

To assay the effects of metal ions and chemical reagents on enzyme activity, fourteen metal ions (K^+^, Na^+^, Cu^2+^, Fe^2+^, Fe^3+^, Mg^2+^, Zn^2+^, Ca^2+^, Pb^2+^, Ni^2+^, Li^+^, Co^2+^, Ag^+^, and Cd^2+^) were prepared with sodium citrate buffer (pH 5.0) at a final concentration of 5 mM, and chemical reagents (SDS, EDTA, methanol, isopropanol, β-mercaptoethanol, and ethanol) were at a final concentration of 5% (*v*/*v*). In the control group, metal ions and chemical reagents were substituted with equal amounts of sodium citrate buffer. Enzyme activity was measured as described above.

### 2.8. Effect of Glucose on Enzyme Activity

The effect of glucose on enzyme activity was assayed at concentrations of glucose from 0 to 400 mM with *p*NPG as the substrate at 100 °C in sodium citrate buffer (0.2 M Na_2_HPO_4_, 0.1 M citric acid, pH 5.0). In the control group, glucose was substituted with an equal amount of sodium citrate buffer, and the reaction value was taken as the maximum relative activity. In addition, with two concentrations of *p*NPG (4 mM and 8 mM) as the substrates, the glucose inhibition constant (*Ki*) was determined by fitting to Dixon plots [29] using the enzyme solution and different concentrations of glucose solutions (0 to 400 mM).

### 2.9. Substrate Specificity and Kinetic Parameters

To investigate the substrate specificity of BglAc, the substrates *p*NPX, *p*NPGal, *p*NPAf, *p*NPman, *o*NPGal, cellobiose, sophorise, and gentiobiose (4 mM) were used to assay enzymatic activity under optimal conditions. The release of *p*NP (at 405 nm) and *o*NP (at 420 nm) were measured by a UV spectrophotometer. The kinetic parameters of BglAc were determined by measuring enzyme activity at various concentrations of *p*NPG (from 0.2 to 2 mM). The reaction was performed in sodium citrate buffer (pH 5.0) at 100 °C for 5 min. The values of *K*_m_, *V*_max_, *K*_cat_, and *K*_cat_/*K*_m_ were determined by fitting the Michaelis–Menten equation.

### 2.10. HPLC Analysis of Soybean Isoflavone Glycosides Hydrolyzed by BglAc

To evaluate the potential application of BglAc, soybean isoflavone glycosides (daidzin, glycitin, and genistin) were used as substrates for biotransformation. The enzymatic hydrolysis reaction took place in a solution with a volume of 3200 μL, of which 800 μL was purified enzyme solution with a final concentration of 27.54 U, 800 μL was a 0.25 mg/mL solution of soybean isoflavone glycosides substrate, and 1600 μL of sterile water were taken. The reaction was carried out at 90 °C and pH 5.0 for 15 min, and the reaction was terminated on ice immediately after the reaction was completed. The purified enzyme solution in the control group was filled with sterile water, and three controls were conducted for each treatment. Finally, the reaction system was transferred to the ultrafiltration tube at 4 °C, 12,000 r/min, 30 min for ultrafiltration. After ultrafiltration, the filtrate was transferred to the sample bottle for HPLC detection and analysis.

The hydrolysis products of daidzin, genistin, and glycitin by BglAc β-glucosidase were assayed using a Shimadzu VP-ODS C18 column (250 mm × 4.6 mm, 5 μm) on an HPLC system equipped with a UV detector set at 254 nm. The column was eluted with a gradient of solvent A (0.1% acetic acid in water) and solvent B (acetonitrile solution) at 25 °C. A total of 5 μL of sample was injected and then solvent B was run at 20% for 5 min, increased from 20 to 60% for 25 min, and increased continuously from 60 to 95% for 10 min at a flow rate of 1.0 mL/min. Then, the peaks of daidzin, glycitin, and genistin, as well as their corresponding isoflavone aglycones, were identified by comparing their elution profile with those of the commercial standards [30,31,32].

### 2.11. Enzyme Characteristics of Mutant BglAcN

The mutant BglAcN, which was lacking a 30 bp nucleotide sequence (the deletion sequence: 5′-CTCAAGGAGCTCGCCGAGGGACGACCTGAG-3′) compared with the wild-type gene *BglAc*, was synthesized. The fragment of the mutant gene was inserted into pET-30a using the same plasmid construct strategy as pET-30a-*BglAc*, described previously. The above-described operations including inducing expression, purification, and relative enzyme activity determination under various conditions for pET-30a-*BglAc* were repeated in an identical form for the mutant, except that the optimal pH, temperature, and thermostability were determined by measuring the relative enzyme activities of the crude enzyme of BglAcN. The same temperature was used to compare the thermostability of wild-type BglAc and the mutant enzyme BglAcN.

## 3. Results

### 3.1. Gene Sequence Analysis

A putative β-glucosidase (BglAc) from *Acidilobus* sp., which was sampled from a hot spring in Yellowstone National Park, USA, was discovered by mining metagenome data in NCBI databases (accession number: WYEC01000097). BglAc is composed of a 1458 bp nucleic acid and encodes 485 amino acid residues. The molecular mass and pI of BglAc were predicted to be 55.3 kDa and 6.65, respectively. The results of protein blast search in the NCBI database demonstrated that BglAc shared high identity similarity with some GH1 family genes, including 4HA3 [24], 1UWQ, 1QVB [33], 3APG [34], and other proteins (Figure 1). Of them, 4HA3 from *A. saccharovorans* 345-15 [24] had the highest identity similarity (81.89%) to BglAc (Table 1). Therefore, the 3D structure of BglAc was predicted using the Swiss-Model server (https://swissmodel.expasy.org/ 6 January 2024) with 4HA3 as the model protein (Figure 2). The sites Glu208 and Glu383 were predicted to be the catalytic sites of BglAc (Figure 2). Furthermore, the multiple sequence alignment analysis showed that there are five binding sites, namely Gln19, His152, Asn207, Glu428, and Trp429 (Figure 1).

### 3.2. Enzyme Expression and Purification

The synthetic sequence of the gene *BglAc* was cloned into the prokaryotic expression vector pET-30a and fused with a 6× His tag at the N-terminus. Then the sequences were expressed as proteins in *E. coli* BL21(DE3) strain with IPTG induction. After IPTG induction and cell ultrasonic disruption, the recombined enzyme was purified with the Ni-NTA column. The protein purification of BglAc was observed on the SDS-PAGE gel (Figure 3), and the molecular weight was approximately 55.3 kDa (not including the his-tag), which was consistent with the theoretical molecular weight.

### 3.3. Effects of pH and Temperature on BglAc and Thermostability Evaluation

With *p*NPG as the substrate, the purified recombinant enzyme BglAc displayed high activity levels (>80%) in the pH range of 4.5–6.0, and the optimum pH was 5.0 (Figure 4A). Under the optimum pH, the enzymatic activity was the highest at 100 °C and remained highly active at 90–100 °C (Figure 4B). After dilution with buffers of different pH levels, the purified enzyme was placed at 80 °C and 100 °C for 1 h, and the results exhibited that the recombinant enzyme BglAc had poor pH stability (Figure 4C). Furthermore, the thermostability of BglAc was also assayed. The purified recombinant proteins of BglAc were preheated in a water bath kettle and were sampled every hour under 90 °C and every ten minutes under 100 °C. Then, the enzyme activity assay was performed. The recombinant β-glucosidase enzyme BglAc showed excellent thermostability under the 90 °C high-temperature treatment, and the relative activity was maintained at almost 70% and 50% at 4 h and 6 h, respectively. In contrast, the relative activity under 100 °C decreased quickly, for example, the relative activity was one-half of that of the untreated enzyme. The half-life values of BglAc at 100 °C and 90 °C were 8 min and 6 h, respectively (Figure 4D).

### 3.4. Effects of Metal Ions and Chemical Reagents on BglAc

The effects of metal ions and reagents on the enzymatic activity of BglAc were investigated with *p*NPG as the substrate at pH 5.0 and 100 °C (Table 1). The activity of the enzymes assayed under the optimum conditions in the absence of additives was defined as 100%, and the relativity activity was presented as a percentage of the optimum activity. The results demonstrated that the activity of BglAc was stimulated by the addition of several metal ions, and an activation of more than 14% was detected in the presence of 5 mM K^+^, Na^+^, Cu^2+^, Fe^3+^, Mg^2+^, and Ca^2+^. Zn^2+^, Ni^2+^, Co^2+^, and isopropanol additions just slightly elevated the enzyme activity by less than 8%. In contrast, the metal ions Li^+^, Fe^2+^, and Pb^2+^ as well as reagents like EDTA, methanol, and β-mercaptoethanol inhibited enzymatic activity, and the activity reduction was limited to 15%. Remarkably, the enzymatic activity was only 20% after adding Cd^2+^ to the reaction system. Similarly, the individual presence of Ag^+^ and SDS inhibited enzymatic activity, and no enzymatic activity was detected. Nonetheless, ethanol with a low concentration addition of 5% increased the enzymatic activity of BglAc (Figure 5A).

### 3.5. Ethanol and Glucose Tolerance of BglAc

In order to test the ethanol tolerance of BglAc under the optimal reaction conditions, the final volumes of 5%, 10%, 20%, and 30% ethanol were added to the reaction system, and the effect of different concentrations of ethanol on BglAc enzyme activity was studied with no ethanol as the control. As shown in Figure 5A, a low concentration of ethanol (<5%) can maintain and slightly increase the enzyme activity. With the increase in ethanol concentration, the enzyme activity gradually decreased. When the ethanol concentration was 25%, the relative enzyme activity was still about half. The effects of glucose on the hydrolysis of *p*NPG by BglAc were investigated using two concentrations of *p*NPG (4 mM and 8 mM) as the substrate. The dosages of 50 mM, 100 mM, and 200 mM of glucose decreased the enzymatic activities of BglAc to 80%, 69%, and 53%, respectively. In addition, with two concentrations of *p*NPG as the substrate, the results of the Dixon plot showed that the *Ki* of BglAc was 180 mM (Figure 5B).

### 3.6. Substrate Specificity and Kinetic Parameters of BglAc

Different substrates that β-glycosides may act on were used to detect substrate specificity. The results showed that BglAc had different levels of hydrolytic capacity for aryl β-glycosides and disaccharide substrates (Table 2). Using *p*NPG as a substrate, the kinetic parameters showed that the *K*_m_, *V*_max_, *K*_cat_, and *K*_cat_/*K*_m_ of BglAc were 3.41 mM, 474.0 μmol min^−1^ mg^−1^, 417.9 s^−1^, and 122.7 s^−1^ mM^−1^, respectively (Table 3), and the specific activity was 357.62 U mg^−1^. Aside from substrate *p*NPG, weak hydrolytic activities were detected for most other substrates under examination, but no activities were detected for the substrates *p*NPGal and *o*NPGal.

### 3.7. Hydrolysis Products of Soybean Isoflavone Glycosides Generated by BglAc

The hydrolysis of isoflavone glycosides by BglAc was determined by HPLC. As shown in Figure 6, soybean isoflavone glycosides daidzin, genistin, and glycitin were completely converted to free aglycones daidzein, genistein, and glycitein after hydrolysis by high-temperature enzyme BglAc at 90 °C and 15 min, and the conversion efficiency could reach 100% compared with the control group. It was found that during the high-temperature hydrolysis experiment, although the high-temperature enzyme also hydrolyzed 100% of the soybean isoflavone glycosides substrate compared to the low-temperature enzyme, the corresponding aglycones (daidzein, glycitein and genistein) product yields were only 77.66%, 73.33% and 53.84%, respectively. Interestingly, this phenomenon was not observed in the control group.

### 3.8. The Optimal Temperature, pH, and Thermostability of Mutant Enzyme BglAcN

To identify amino acids that contribute to the thermophilic and thermal stability of BglAc, mutant BglAcN was constructed and expressed by IPTG. Unlike the wild-type BglAc, the expression level of the mutant enzyme protein is low, and it is difficult to obtain sufficient purification for subsequent enzyme activity determination. Therefore, we had to compare the crude enzyme solution of BglAcN with the wild-type enzyme. After dilution to a suitable concentration, the crude enzymes of BglAc and BglAcN were used to measure enzyme activity. Results showed that the optimal pH was not changed between BglAc and the mutant, but the optimal temperature of BglAcN was 90 °C and lower than that of BglAc (Figure 7A,B). An interesting observation was that the relative enzyme activities of BglAcN decreased to a very low level (10%) and seemed to be activated by high temperature (Figure 7C). According to sequence alignment and three-dimensional structure analysis, the differences between BglAc and 3APG from typical thermophilic bacteria *P. furiosus* mainly existed in the 320–342 and 360–375 regions. However, there is a significant difference in thermal stability between them, which may be due to oligomerization driven by several subunits, the accumulation of ion pairs and hydrogen bonds on the protein surface, increasing hydrophobicity and packing density in the protein core, and an entropic effect caused by shortened surface loops or the introduction of proline residues into loops [35].

## 4. Discussion

The determination of the crystal structures of β-glucosidase from hyperthermophilic organisms such as *P. furiosus* [13,19,34], *S. solfataricus* [20], *T. acidophilum* [22], *S. acidocaldarius* [23], *A. saccharovorans* [24], and *T. kodakarensis* [26] have supplied useful information to be investigated in further studies. In this report, the gene *BglAc* was amplified from *Acidilobus* sp. that belongs to a thermophilic genus in a sample isolated from a hot spring in Yellowstone National Park in the USA. The natural habitat of these living organisms is an environment with high temperature and acidic conditions. Indeed, the optimal pH and optimal temperature for BglAc activity are consistent with these environmental conditions. According to previous studies, β-glucosidases of most archaea possess neutral to acidic and hyperthermophilic characteristics. Similar to the β-glucosidases of archaea that have been investigated previously, the optimum pH and temperature of BglAc are pH 5.0 and 100 °C. These values of the temperature and pH optima of BglAc are typical for β-galactosidases and β-glucosidases from hyperthermophilic archaea, including enzymes from *S. solfataricus* (85 °C and pH 5.5 [20]), *P. furiosus* (100 °C and pH 5.0 [19]), *S. acidocaldarius* (90 °C and pH 5.5 [23]), and *Thermococcus kodakarensis* (105 °C and pH 7.0 [26]) (Table 4). In terms of thermal inactivation, BglAc is one of the most thermostable β-glucosidases with a half-life of 6 h at 90 °C. It shows a higher stability than β-glucosidases from *Alicyclobacillus* sp. [16] and *T. acidophilum* but a lower stability than those from *P. furiosus* (85 h at 100 °C) and *S. solfataricus* (48 h at 85 °C).

Most of the reports on β-glucosidases suggest that their optimum activity levels occur at a temperature range of 40–70 °C and a pH range of 5–8 [1]. Within these ranges, the optimum temperature of low-temperature enzymes is concentrated at 30–40 °C, and the optimum temperature of medium-temperature enzymes is concentrated at 50–70 °C. Compared with mesophilic enzymes from hot springs [36] and *Jeotgalibacillus malaysiensis* [37] and low-temperature enzymes from *Bacillus cellulosilyticus* [38] and *Exiguobacterium antarcticum* B7 [39], BglAc has better thermal stability under high-temperature conditions, which provides greater possibilities for its application in high-temperature industries.

**Table 4 microorganisms-12-00533-t004:** Enzymatic activities of some thermostable β-glucosidases that are publicly available.

Organisms	Enzyme	Optimum Temperature (°C)	Optimum pH	T _1/2_	Reference
*Alicyclobacillus* sp.	AsBG1	100	6.5	1 h (50 °C)	[16]
*P. furiosus*	/	100	5.0	85 h (100 °C)	[19]
*S. solfataricus*	Sβ-gly	85	5.5	48 h (85 °C)	[20]
*P. kodakaraensis*	Pk-gly	100	6.5	18 h (90 °C)	[21]
*T. acidophilum*	TaBglA	90	6.0	0.4 h (90 °C)	[22]
*S. acidocaldarius*	bgaS	90	5.5	0.2 h (90 °C)	[23]
*A. saccharovorans*	Asac_1390	93	6.0	7 h (90 °C)	[24]
*T. naphthophila*	BglA	95	7.0	8 h (80 °C)	[25]
*T. kodakarensis*	vul_bgl1A	105	7.0	0.25 h (90 °C)	[26]
Hot spring	bgl_M_	70	5.0	32 h (60 °C)	[36]
*J. malaysiensis*	BglD5	65	7.0	0.6 h (65 °C)	[37]
*B. cellulosilyticus*	BcBgl1A	40	7.0	24 h (40 °C)	[38]
*E. antarcticum* B7	EaBgl1A	30	7.0	48 h (30 °C)	[39]
*Thermococcus* sp.	O08324	78	6.5	14.3 h (78 °C)	[40]
*Acidilobus* sp.	BglAc	100	5.0	6 h (90 °C)	This study

Broad substrate specificity and relatively high catalytic efficiency provide β-glucosidases with many potential application prospects, such as feed, bioenergy, food, textile, and other fields [1,2,8]. The typical thermophilic β-glucosidases which come from *P. furiosus* (*p*N*P*G > *p*NPGal > *p*NPX > *p*NPman) [19], *S. solfataricus* (*p*N*P*G > *p*NPAf > *o*NPGal > *p*N*P*Gal) [20], and *A. saccharovorans* (*p*N*P*Gal > *o*NPGal > *p*NPG > *p*NPX > *p*NPman) [24] all have broad substrate specificity. In addition to their ability to hydrolyze aromatic substrates, they also have varying degrees of ability to hydrolyze cellobiose, as does BglAc. Interestingly, although BglAc has different degrees of hydrolysis ability for aryl β-glucosidase and disaccharide substrates, it has only weak hydrolysis activity for other aromatic substrates except for the substrate *p*NPG, and hydrolysis activity was not detected at all in the substrates *p*NPGal and *o*NPGal.

Mutant BglAcN was formed through the deletion of nine amino acids (SRSSPRDDL) at positions 298 to 307 compared to the wild-type enzyme BglAc. The mutant strategy is based on the analysis of multiple alignments of homologous genes encoding β-glucosidase from bacteria, archaea, and fungi. The relative activity of mutant BglAcN was very low compared to that of wild-type BglAc, and it is difficult to obtain enough purified protein to determine the enzyme characteristics. The difference in fold numbers between BglAc and BglAcN was up to 500–1000 folds when we measured the crude enzyme activity. So, it seems that the amino acid residues of SRSSPRDDL are vital for the expression of BglAc and β-glucosidase activity. The *K*_m_ values of mutant BglAcN are higher than those of wild type β-glucosidase, while specific activity, *V*_max_, *K*_cat_, and *K*_cat_/*K*_m_ were reduced. It seems that catalysis in mutant BglAcN is achieved at the expense of substrate affinity. The deletion of amino acid residues between 298 to 307 locations in BglAc consist of a β-fold (seventh β-fold) in the β-glucosidase secondary structure, and these amino acid residues are near one catalysis site (Glu383) in the spatial structure. So, it is reasonable that the catalysis efficiency was reduced in the deletion mutant. This result indicates that this deleted section is important for β-glucosidase thermostability and is consistent with the initial assumption that this section exists in BglAc and 4H3A (*A. saccharovorans* [24]) but not in 3APG (*P. furiosus* [34]), which probably contributes to the ability to sustain activities under higher temperatures for a long time. Considering the relationship between mutant sites and the changes that occurred in enzyme characteristics, some emphasis is placed on the need for future studies.

Hyperthermophilic microorganisms are attractive in industry applications because of their thermostability in high-temperature fermentation environments and resistance to organic solvents that are often used in fermentation. For example, β-glucosidase has been identified as a key enzyme in bioethanol production and active aglycone hydrolysis, which represents the main bottleneck in efficient hydrolysis [41]. In addition, the enzymatic hydrolysis of β-glucosidase to prepare soybean isoflavone glycosides is an important application with high commercial value [42], which has been investigated through the direct use of β-glucosidases from leguminous plants and microorganisms [3,9,15]. However, in the industrial production of isoflavones, the low solubility of isoflavones, the poor stability of β-glucosidase, product inhibition, limited enzymatic hydrolysis efficiency, and other scientific problems reduce the conversion efficiency of isoflavone aglycones [1,18,43]. In view of these issues, the hydrolysis rate of soybean isoflavones by recombinant β-glucosidase BglAc was determined using soybean isoflavone glycosides (daidzin, glycitin, and genistin) as substrates. BglAc showed the highest hydrolysis activity for soybean isoflavone glycosides after treatment at 90 °C for 15 min, and the hydrolysis rate could reach almost 100%. It was found that during the high-temperature hydrolysis experiment, although the high-temperature enzyme also hydrolyzed 100% of the soybean isoflavone glycosides (daidzin, glycitin, and genistin) substrates compared to the low-temperature enzyme, the corresponding aglycones (daidzein, glycitein, and genistein) product yields were only 77.66%, 73.33%, and 53.84%, respectively.

## 5. Conclusions

In this study, β-glucosidase of *Acidilobus* sp. belonging to family GH1 was expressed and characterized by mining metagenome data sampling from a hot spring located at Yellowstone National Park in the USA. The optimal pH and temperature of BglAc are 5.0 and 100 °C, respectively. The enzyme is still active at high temperature (90 °C) for 6 h and displays exclusively hydrolytic activity with a specific activity of 357.62 U mg^−1^. The tolerance to high temperature, acidic conditions, and organic solvent ethanol makes the enzyme promising for applications in specific industrial degradation processes such as the hydrolysis of soybean isoflavone glycosides.

## Figures and Tables

**Figure 1 microorganisms-12-00533-f001:**
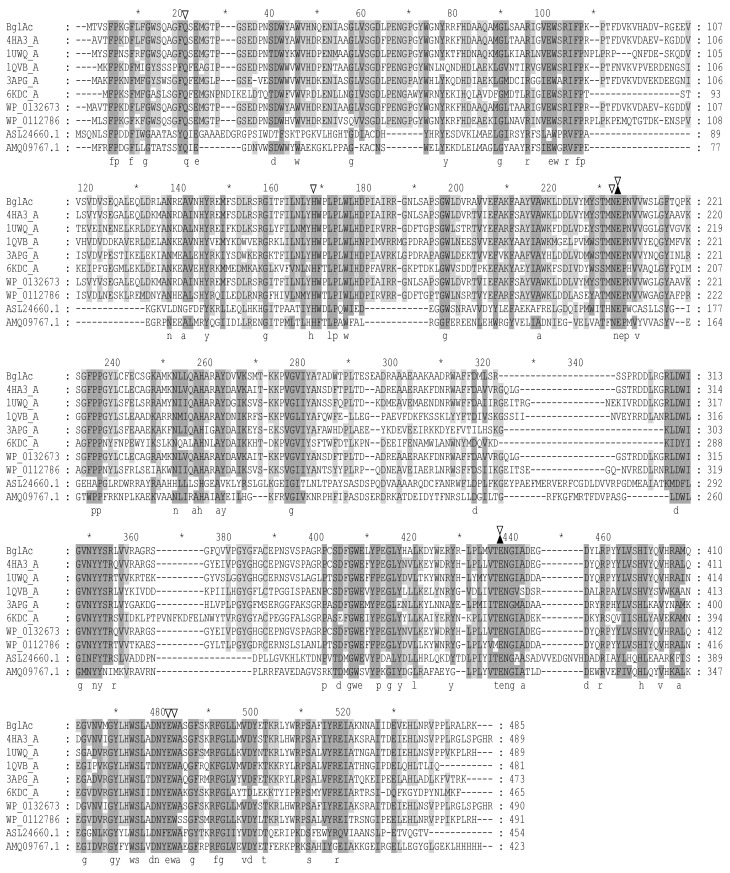
Multiple sequence alignment of BglAc. Aside from BglAc, all protein sequences were obtained from GenBank (https://www.ncbi.nlm.nih.gov/ 6 January 2024). The aligned sequences originated from the following organisms: 4HA3_A, *A. saccharovorans*; 1UWQ_A, *Saccharolobus solfataricus*; 1QVB_A, *Thermosphaera aggregans*; 3APG_A, *Pyrococcus furiosus*; 6KDC_A, *Fervidobacterium pennivorans*; WP_013267307.1, *A. saccharovorans*; WP_011278657.1, *S. acidocaldarius*; ASL24660.1, *Alicyclobacillus* sp.; AMQ09767.1, *Thermococcus* sp. The alignment was performed by the Clustal W method and annotated by GeneDoc. *: The catalytic site of the enzyme. The two catalytic sites are indicated by a solid triangle (▲), and the binding sites are hollow inverted triangles (▽). Identical amino acids are shown on a black background, similar amino acids are shown on a gray background, and different amino acids are shown on a white background.

**Figure 2 microorganisms-12-00533-f002:**
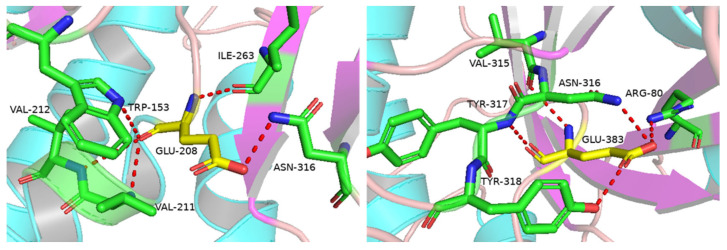
Catalytic sites in the structure of BglAc. The catalytic sites are colored yellow; the residues are green. The 3D structure of BglAc was predicted using the Swiss-Model server (https://swissmodel.expasy.org/ 6 January 2024) with 4HA3_A as the model, and the pictures were drawn with Pymol (https://pymol.org/2/ 6 January 2024).

**Figure 3 microorganisms-12-00533-f003:**
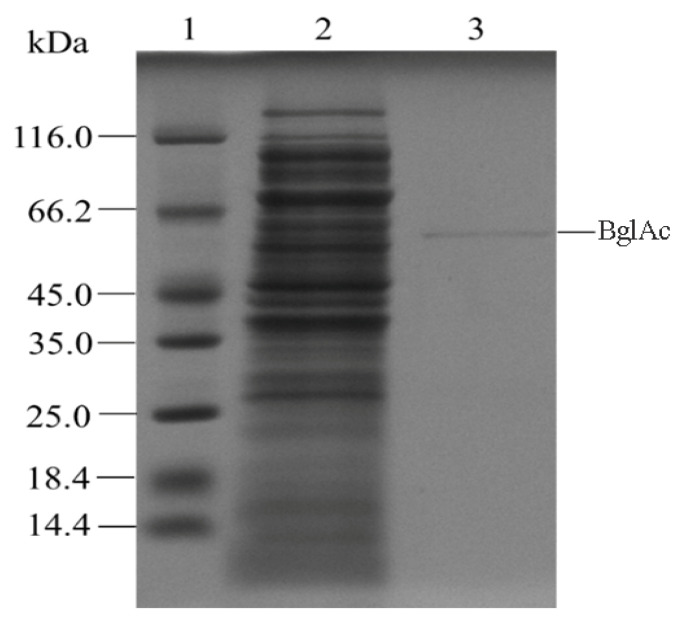
Expression and purification of recombinant glucosidase BglAc. Lanes: 1—molecular weight markers (size are shown in kDa); 2—total protein from induced cells; 3—purified recombinant BglAc.

**Figure 4 microorganisms-12-00533-f004:**
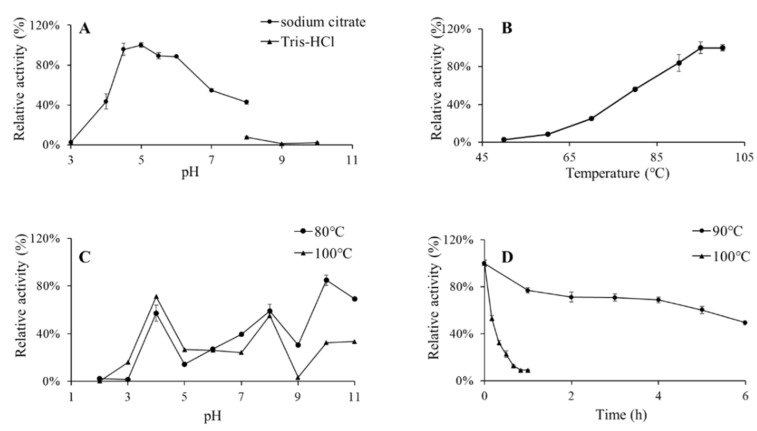
Effects of pH and temperature on the β-glucosidase activity of BglAc. (**A**) Effect of pH on the BglAc activity. The used buffers were 50 mM sodium citrate buffer in the range of pH 3.0–8.0 (filled circles) and 50 mM sodium phosphate buffer in the range of pH 8.0–10.0 (filled triangle). (**B**) Effect of temperature on the BglAc activity. (**C**) pH stability. (**D**) Thermostability. Data represent the means of triplicate measurements and error bars represent standard deviation.

**Figure 5 microorganisms-12-00533-f005:**
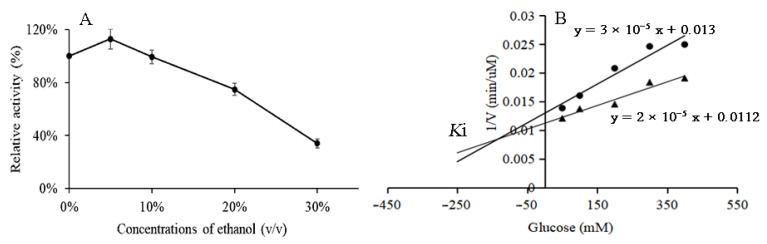
Dixon plot and ethanol tolerance diagram of the inhibition of glucose on *p*NPG hydrolysis activity of BglAc. (**A**) Effect of ethanol on the activity of BglAc. (**B**) Effect of glucose on the activity of BglAc. Black dots (•) and solid triangle (▲) represent 4 mM and 8 mM *p*NPG. Data are means of triplicates, and the error bars show standard deviations (mean ± SD).

**Figure 6 microorganisms-12-00533-f006:**
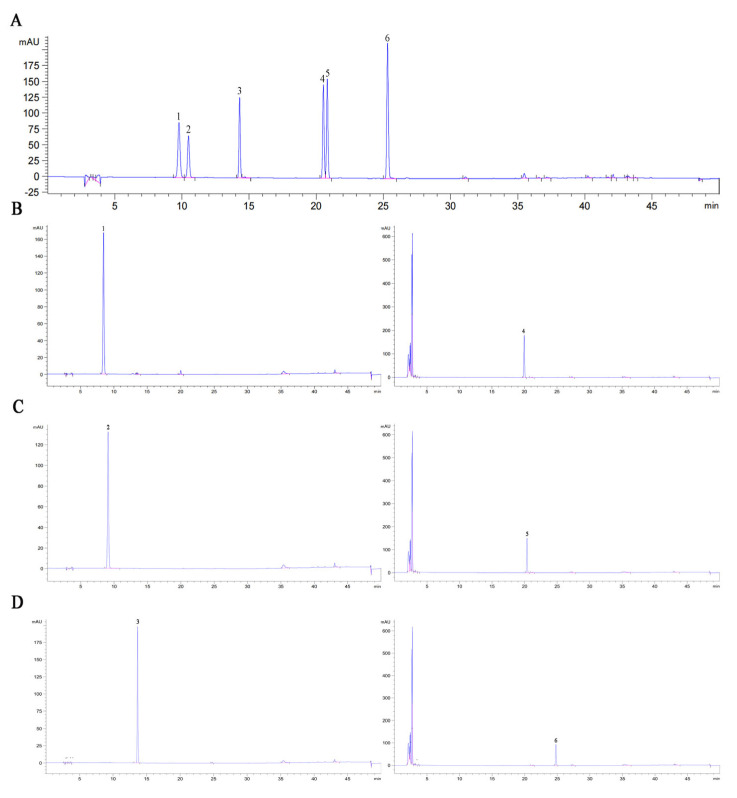
HPLC of soybean isoflavone standards and BglAc hydrolyzed soybean isoflavone glycosides. (**A**) HPLC profiles of daidzin, glycitin, genistin, daidzein, glycitein, and genistein. (**B**) HPLC detection of soybean isoflavone glycoside daidzin hydrolyzed by BglAc. (**C**) HPLC detection of soybean isoflavone glycoside glycitin hydrolyzed by BglAc. (**D**) HPLC detection of soybean isoflavone glycoside genistin hydrolyzed by BglAc. Notes: 1—daidzin (9.781 min); 2—glycitin (10.490 min); 3—genistin (14.298 min); 4—daidzein (20.531 min); 5—glycitein (20.821 min); 6—genistein (25.311 min).

**Figure 7 microorganisms-12-00533-f007:**
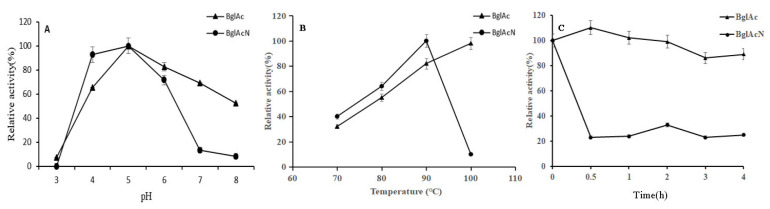
Enzymatic characteristics of wild-type BglAc and mutant BglAcN. (**A**) Effect of mutation sites on optimal pH; (**B**) effect of mutation sites on optimal temperature; (**C**) effect of mutation sites on thermostability. The enzyme activities were assayed by the crude enzyme.

**Table 1 microorganisms-12-00533-t001:** Effects of metal ions and chemical additives on the activity of BglAc.

Reagent	Relativity Activity (%)	Reagent	Relativity Activity (%)
	5 mM		5 mM
Control ^a^	100.0 ± 5.72	Ni^2+^	103.5 ± 7.35
K^+^	116.7 ± 0.72	Li^+^	95.6 ± 9.56
Na^+^	114.0 ± 9.65	Co^2+^	104.7 + 8.97
Cu^2+^	116.9 ± 4.20	Ag^+^	ND
Fe^2+^	85.0 ± 5.32	Cd^2+^	20.0 ± 1.03
Fe^3+^	121.7 ± 1.06	SDS	ND
Mg^2+^	124.4 ± 6.35	EDTA	87.2 ± 5.31
Zn^2+^	103.5 ± 9.42	methanol	89.1 ± 0.35
Ca^2+^	118.6 ± 0.49	isopropanol	107.6 ± 1.6
Pb^2+^	72.4 ± 4.24	β-mercaptoethanol	64.1 ± 3.96

Data are presented as means ± SD. ^a^: In the control group, metal ions and chemical reagents were substituted with equal amounts of sodium citrate buffer.

**Table 2 microorganisms-12-00533-t002:** Substrate specificity of BglAc.

Substrate	Specific Activity (U mg^−1^)
Disaccharide	
Cellobiose	ND
Sophorise	1.11 ± 0.22
Gentiobiose	14.93 ± 0.90
Aryl β-glycoside	
*p*NPG	357.62 ± 1.85
*p*NPX	2.93 ± 0.20
*p*NPGal	ND
*p*NPAf	2.90 ± 0.07
*p*NPman	1.78 ± 0.39
*o*NPGal	ND

Data represent the means of three separate replications. ND means no activity was detected by the method used in this study.

**Table 3 microorganisms-12-00533-t003:** Kinetic parameters of wild and mutant β-glucosidase (*p*NPG as substrate).

Enzyme	Specific Activity (U mg^−1^)	*K*_m_ (mM)	*V*_max_ (μmol min^−1^ mg^−1^)	*K*_cat_ (s^−1^)	*K*_cat_/*K*_m_ (s^−1^ mM^−1^)
BglAc (WT)	357.62	3.41	474.0	417.9	122.7
BglAcN (Mutant)	52.44	8.58	287.87	259.66	30.27

Data were measured with a purification enzyme for BglAc and a crude enzyme for BglAcN.

## Data Availability

Data are contained within the article.

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
