# Peer review of "Characterization of a Novel Hyperthermophilic GH1 β-Glucosidase from Acidilobus sp. and Its Application in the Hydrolysis of Soybean Isoflavone Glycosides"

_microorganisms, 2024, doi:10.3390/microorganisms12030533_

Round 1
Reviewer 1 Report
Comments and Suggestions for Authors
This paper presents the characterization of a novel hyperthermophilic GH1 β-Glucosidase from Acidilobus sp., named BglAc, and its application in the hydrolysis of soybean isoflavone glycosides. BglAc, identified from a hot spring in Yellowstone Park, shows exceptional thermostability and ethanol tolerance, with optimal activity at pH 5.0 and 100°C. The enzyme retains significant activity even in the presence of high glucose and ethanol concentrations, making it a promising candidate for applications in the food industry, animal feed, and biomass degradation. Before its publication, the following issues should be considered:
1. Comparative Analysis: The paper could have benefited from a more extensive comparison with other known β-glucosidases to highlight the unique advantages of BglAc.
2. Mechanistic Insights: While the enzyme's functional characteristics are well-described, insights into the molecular basis of its thermostability and ethanol tolerance are lacking.
3. Application Scope: The focus is mainly on soybean isoflavone glycosides; exploring a broader range of substrates could enhance the enzyme's utility across different industrial sectors.
Comments on the Quality of English LanguageThis paper presents the characterization of a novel hyperthermophilic GH1 β-Glucosidase from Acidilobus sp., named BglAc, and its application in the hydrolysis of soybean isoflavone glycosides. BglAc, identified from a hot spring in Yellowstone Park, shows exceptional thermostability and ethanol tolerance, with optimal activity at pH 5.0 and 100°C. The enzyme retains significant activity even in the presence of high glucose and ethanol concentrations, making it a promising candidate for applications in the food industry, animal feed, and biomass degradation. Before its publication, the following issues should be considered:
1. Comparative Analysis: The paper could have benefited from a more extensive comparison with other known β-glucosidases to highlight the unique advantages of BglAc.
2. Mechanistic Insights: While the enzyme's functional characteristics are well-described, insights into the molecular basis of its thermostability and ethanol tolerance are lacking.
3. Application Scope: The focus is mainly on soybean isoflavone glycosides; exploring a broader range of substrates could enhance the enzyme's utility across different industrial sectors.
Reviewer 2 Report
Comments and Suggestions for Authors The authors set the stage to investigate a putative β-glucosidase gene, BglAc, which was amplified from Acidilobus sp. in metagenome database sampling from the hot spring of Yellow Stone Park. The gene was was successfully expressed in Escherichia coli with a molecular weight of approximately 55.3 kDa. The BglAc showed the maximum activity using p-nitrophenyl-β-D-glucopyranoside (pNPG) as substrate at optimal pH 5.0 and 100 °C and exhibited extraordinary thermostability at 90 °C. Moreover, a significantly ethanol tolerance, retaining 96% relative activity at 10% ethanol, even 78% at 20% ethanol, enrolled BglAc as a promising enzyme for cellulose saccharification. The authors claim that BglAc is actually a new β-glucosidase with excellent thermostability, ethanol tolerance and glycoside hydrolysis ability, indicating its good application prospects in the food, animal feed, and lignocellulosic biomass degradation. The study is interesting but there are some issues that have to be addressed. 1) The authors studied the effect of metal ions and reagents on the enzymatic activity of BglAc with pNPG as a substrate at pH 5.0 and 100 oC. Have the authors tested the effect of metal ions at 90 oC, considering that the enzyme performed excellent thermostability and good relative activity under 90oC ? 2) What would be the effect of ethanol at the two different concentrations (4mM and 8mM) of pNPG (Present it to the diagram 5A, as in diagram 5B) ?Author Response
Please see the attachment.

Round 2
Reviewer 1 Report
Comments and Suggestions for Authors
Accept
Comments on the Quality of English LanguageAccept
Reviewer 2 Report
Comments and Suggestions for Authors
-